# Dynamics of the cell fate specifications during female gametophyte development in *Arabidopsis*

**Daichi Susaki**[1], **Takamasa Suzuki**[2], **Daisuke Maruyama**[1], **Minako Ueda**[3,4¤],
**Tetsuya Higashiyama**[3,4,5]*, **Daisuke Kurihara**[3,6]*

**1** Kihara Institute for Biological Research, Yokohama City University, Yokohama, Japan, **2** Department of
Biological Chemistry, College of Bioscience and Biotechnology, Chubu University, Kasugai, Japan, **3** Institute
of Transformative Bio-Molecules (ITbM), Nagoya University, Nagoya, Japan, **4** Division of Biological Science,
Graduate School of Science, Nagoya University, Nagoya, Japan, **5** Department of Biological Sciences,
Graduate School of Science, University of Tokyo, Tokyo, Japan, **6** JST, PRESTO, Nagoya, Japan

¤ Current address: Graduate School of Life Sciences, Tohoku University, Sendai, Japan
* higashi@bio.nagoya-u.ac.jp (TH); kuri@bio.nagoya-u.ac.jp (DK)

journal.pbio.3001123

STATES

**Data Availability Statement:** RNA-seq data
associated with this study have been deposited in
DDBJ Sequence Read Archive (DRA) under the
accession number, DRR220104–DRR220111.

## Abstract

The female gametophytes of angiosperms contain cells with distinct functions, such as
those that enable reproduction via pollen tube attraction and fertilization. Although the
female gametophyte undergoes unique developmental processes, such as several rounds
of nuclear division without cell plate formation and final cellularization, it remains unknown
when and how the cell fate is determined during development. Here, we visualized the living
dynamics of female gametophyte development and performed transcriptome analysis of
individual cell types to assess the cell fate specifications in *Arabidopsis thaliana*. We
recorded time lapses of the nuclear dynamics and cell plate formation from the 1-nucleate
stage to the 7-cell stage after cellularization using an in vitro ovule culture system. The mov-
ies showed that the nuclear division occurred along the micropylar–chalazal (distal–proxi-
mal) axis. During cellularization, the polar nuclei migrated while associating with the forming
edge of the cell plate, and then, migrated toward each other to fuse linearly. We also tracked
the gene expression dynamics and identified that the expression of *MYB98pro*::*GFP–
MYB98*, a synergid-specific marker, was initiated just after cellularization in the synergid,
egg, and central cells and was then restricted to the synergid cells. This indicated that cell
fates are determined immediately after cellularization. Transcriptome analysis of the female
gametophyte cells of the wild-type and *myb98* mutant revealed that the *myb98* synergid
cells had egg cell–like gene expression profiles. Although in *myb98*, egg cell–specific gene
expression was properly initiated in the egg cells only after cellularization, but subsequently
expressed ectopically in one of the 2 synergid cells. These results, together with the various
initiation timings of the egg cell–specific genes, suggest complex regulation of the individual
gametophyte cells, such as cellularization-triggered fate initiation, MYB98-dependent fate
maintenance, cell morphogenesis, and organelle positioning. Our system of live-cell imag-
ing and cell type–specific gene expression analysis provides insights into the dynamics and
mechanisms of cell fate specifications in the development of female gametophytes in plants.

Public data of egg cell, ovule and seedling were DRR174980, DRR174981, DRR174982, DRR044370, DRR066525, SRR346552, SRR346553. The Supporting information are available in the online version of this article.

**Funding:** The authors have received funding from the followng sources: MEXT | Japan Society for the Promotion of Science (JSPS):JP19H04869 (DM); MEXT | Japan Society for the Promotion of Science (JSPS):JP17H05838, JP19H04859, JP19H05670, JP19H05676, JP19H03243, JP19K22421 (MU); MEXT | Japan Society for the Promotion of Science (JSPS):JP16H06464, JP16H06465 (TH); MEXT | Japan Society for the Promotion of Science (JSPS):JP10J07811, JP18J01963, JP19K16172 (DS); MEXT | Japan Society for the Promotion of Science (JSPS):JP17H03697, JP18K19331, 20H05358 (DK); JST | Precursory Research for Embryonic Science and Technology (PRESTO): JPMJPR18K4 (DK) The funders had no role in study design, data collection and analysis, decision to publish, or preparation of the manuscript.

**Competing interests:** The authors have declared that no competing interests exist.

**Abbreviations:** CCD, charge-coupled device; CDR1, CONSTITUTIVE DISEASE RESISTANCE 1; CRP, cysteine-rich peptide; CTPP, COOH-terminal propeptide; DEG, differentially expressed gene; ER, endoplasmic reticulum; GO, Gene Ontology; HAE, hour after emasculation; PCA, principal component analysis; RNA-seq, RNA sequencing; SP, signal peptide; STEP, sensor for transiently expressed proteins; TPM, transcripts per million.

## Introduction

In multicellular organisms, each differentiated cell creates complex structures to perform specified functions. As cells differentiate according to their cell fate, it is important that cell fate is determined at the appropriate time and position. However, the molecular mechanisms that determine how cells recognize positional information and their cell fates in plants are not well understood. The development of the female gametophyte in angiosperms is an attractive model for studying cell fate specifications.

Female gametophytes in angiosperms contain highly differentiated cells with distinct functions, such as those for pollen tube attraction and fertilization, which enable plant reproduction. In *Arabidopsis thaliana*, 1 megaspore undergoes 3 rounds of mitosis without cytokinesis as a coenocyte. Cellularization occurs almost simultaneously around each nucleus, producing the *Polygonum*-type female gametophyte with 8 nuclei and 7 cells: 1 egg cell, 1 central cell, 2 synergid cells, and 3 antipodal cells. It is important for the sexual reproduction of angiosperms that each cell of the female gametophyte develops by acquiring its appropriate cell fate. Although it remains unknown when and how the cell fate is determined during female gametophyte development, 2 mechanisms are thought to play important roles: cell polarity along the micropyle–chalazal axis in the female gametophyte and cell–cell communications after cellularization. The female gametophytes of angiosperms develop with distinct polarity. In many plant species, the egg and synergid cells form at the micropylar end of the ovule, and antipodal cells form at the opposite side of the chalazal end [1,2].

In flowering plants, 2 female gametes are fertilized by 2 sperm cells carried by the pollen tube. The egg cell is the female gamete that forms the embryo in the seed by fertilization with the sperm cell. The central cell is regarded a gamete because it is also fertilized by the sperm cell, but it forms the embryo-nursing tissue in the endosperm of the seed, and it is not inherited in the next generation. The synergid cell has finger-like plasma membrane invaginations with thickened cell walls termed "filiform apparatus" in the micropylar end. These structures increase the surface area of the synergid cells with a higher rate of exocytosis for secretion. When the pollen tube arrives at the synergid cells, the synergid cells stop the elongation of the pollen tube and cause the release of sperm cells by rupturing its tip [3].

Genes expressed specifically in each female gametophyte cell and used as markers of cell fate have been identified in several plants, particularly *Arabidopsis* [4]. However, it is not clear when and how these cells specify their cell fates and exhibit specific gene expressions. Mutant analysis has shown a strict correlation between nuclear position and cell fate [5–9]. However, it is still unknown if nuclear position determines cell fate, as there is little spatiotemporal information available on the detailed nuclear dynamics and cell fate specifications. As the female gametophyte development occurs deep within the female pistil, observing it directly in its living state has proven challenging. Therefore, the intracellular behavior of female gametophyte development has been analyzed, by fixing the ovules and observing the sections. It is crucial to capture the living dynamics in the female gametophyte development to reveal the dynamics of cell fate specification.

Here, we performed live cell imaging of female gametophyte development in *Arabidopsis* using an in vitro ovule culture system, which enabled us to observe the nuclear dynamics, division, cellularization, and cell fate specifications in real time, by using specific fluorescent marker lines. Subsequently, we established a method for the isolation of each of the female gametophyte cells with high efficiency, without contaminating the other *Arabidopsis* cells. We then performed transcriptome analysis using a next-generation sequencer for a small number of isolated female gametophyte cells. Furthermore, we analyzed the contributions of the cell–cell communications with regard to changing gene expression, by analyzing the expression

profiles of the synergid cells of the *myb98* mutant, a transcription factor that is thought to contribute to the determination of the synergid cell fate.

## Materials and methods

### Plant materials and growth conditions

For all experiments, the *A. thaliana* accession Columbia (Col-0) was used as the wild type. All *A. thaliana* transgenic lines were in a Columbia (Col-0) background, and the *myb98* mutant was previously described [10]. The following transgenic lines were also previously described: *RPS5Apro::H2B–tdTomato* [11], *RPS5Apro::tdTomato–LTI6b* [12], *RPS5Apro::H2B–sGFP* [13], *FGR8.0* [14], *MYB98pro::GFP* [10], *MYB98pro::GFP–MYB98* [15], *EC1.2pro::mtKaede* [16], *FWApro::FWA–GFP* [17], and *ABI4pro::H2B–tdTomato* [18]. The transgenic lines used are listed in S1 Table.

Arabidopsis seeds were sown on plates containing half-strength Murashige and Skoog salts (Duchefa Biochemie, Haarlem, the Netherlands), 0.05% MES-KOH (pH 5.8), 1× Gamborg's vitamin solution (Sigma, St Louis, Missouri, United States of America), and 1% agar. The plates were incubated in a growth chamber at 22°C under continuous lighting after cold treatments at 4°C for 2 to 3 days in the dark. Two-week-old seedlings were transferred to soil and grown at 21 to 25°C under long-day conditions (16-hour light/8-hour dark).

### Plasmid construction

*GPR1pro::H2B–mNeonGreen* (coded as DKv1200) was constructed with the 2,568-bp upstream region of *GPR1* (At3g23860) and the full-length coding region of *H2B* (HTB1: At1g07790) fused to *mNeonGreen* (Allele Biotechnology, San Diego, California, USA) with the $(SGGGG)_2$ linker, and the 1,959-bp downstream regions were cloned into the binary vector pPZP211 [19]. *CDR1–LIKE2pro::CDR1–LIKE2–mClover* (coded as DKv1023) was constructed using the 1,398-bp upstream region and the full-length coding region of *CDR1–LIKE2* (At1g31450) fused to *mClover* with the $(SGGGG)_2$ linker and *NOS* terminator and cloned into the binary vector pPZP211. *CDR1–LIKE1pro::CDR1–LIKE1–mClover* (coded as DKv1024) was constructed using the 2,000-bp upstream region and the full-length coding region of *CDR1–LIKE1* (At2g35615) fused to *mClover* with the $(SGGGG)_2$ linker and *NOS* terminator and then cloned into the binary vector pPZP211. Finally, *CDR1pro::CDR1–mClover* (coded as DKv1025) was constructed with the 1,577-bp upstream region and the full-length coding region of *CDR1* (At5g33340) fused to *mClover* with the $(SGGGG)_2$ linker and *NOS* terminator and then cloned into the binary vector pPZP211.

To construct the multiple cell type–specific marker line with the nuclei marker (coded as DKv1110), the following sequences were cloned into the binary vector pPZP211, and *NPTII* was replaced with *mCherry* under the control of the *At2S3* promoter from a pAlligator-derived binary vector [20]: *EC1.1pro::SP–mTurquoise2–CTPP* [21] (the 463-bp *EC1.1* promoter was fused to *mTurquoise2* that fused to the signal peptide (SP) sequence of *EXGT–A1* (At2g06850) at the N-terminus and to a vacuolar sorting signal COOH-terminal propeptide (CTPP), and the *HSP* terminator); *DD1pro::ermTFP1* (the 1,262-bp *DD1* promoter (At1g36340) was fused to mTFP1 that was fused to the SP sequence of *EXGT–A1* at the N-terminus and to an endoplasmic reticulum (ER) retention signal (HDEL) at the carboxyl terminus, and the *OCS* terminator); *MYB98pro::mRuby3–LTI6b* (the 1,610-bp *MYB98* promoter and *mRuby3* fused to the start codon of *LTI6b* (At3g05890) with the $(SGGGG)_2$ linker, and the *HSP* terminator); and *AKVpro::H2B–mScarlet–I* (the 2,949-bp upstream regions of *AKV* (At4g05440 [22]) and the full-length coding region of *H2B* (HTB1: At1g07790) fused to *mScarlet–I* with the $(SGGGG)_2$ linker). To enhance the expression level of H2B–mScarlet–I, the 5′ UTR of *AtADH* (alcohol

dehydrogenase) was inserted between the *AKV* promoter and *H2B* coding sequence. Unfortunately, H2B–mScarlet–I was expressed not only in female gametophytes but also in sporophytic cells for this construct.

*SBT4.13pro*::*SBT4.13–mClover* (coded as pDM349) was a 2,040-bp upstream region, and the full-length coding region of *SBT4.13* (At5g59120) was amplified and cloned into pPZP221Clo using a SmaI site [23].

*MYB98pro*::*NLS–mRuby2* (coded as pDM371), a DNA fragment of NLS–mRuby2 (obtained from Addgene plasmid 40260), was amplified and then cloned into the pENTR/D-TOPO vector (Invitrogen, Japan) to generate pOR006. LR recombination between pDM286 [13] and pOR006 was performed using LR clonaseII (Invitrogen) to produce pDM371.

The binary vectors were introduced into the *Agrobacterium tumefaciens* strain EHA105. The floral dip or simplified *Agrobacterium*-mediated methods were used for *Arabidopsis* transformations [24]. The transgenic lines and primers are listed in S1 and S2 Tables, respectively.

## Microscopy

To image female gametophyte development, we used 2 spinning disk confocal microscope systems following the settings of [25], with the following modification: For the live imaging of the in vitro female gametophyte development, the confocal images were acquired using an inverted fluorescence microscope (IX-83; Olympus, Tokyo, Japan), equipped with an automatically programmable XY stage (BioPrecision2; Ludl Electronic Products, Hawthorne, New York, USA), a disk-scan confocal system (CSU-W1; Yokogawa Electric, Tokyo, Japan), 488-nm and 561-nm LD lasers (Sapphire; Coherent, Santa Clara, USA), and an EMCCD camera (iXon3 888; Andor Technologies, South Windsor, Connecticut, USA). Time-lapse images were acquired with a 60× silicone oil immersion objective lens (UPLSAPO60XS, WD = 0.30 mm, NA = 1.30; Olympus) mounted on a Piezo focus drive (P-721; Physik Instrumente, Karlsruhe, Germany). We used 2 band-pass filters, 520/35 nm for the GFP and 593/46 nm for the tdTomato. The images were processed with Metamorph (Universal Imaging, Downingtown, USA) and Fiji [26] to create maximum-intensity projection images and to add color.

We also used an inverted confocal microscope system with a stable incubation chamber (CV1000; Yokogawa Electric) equipped with 488-nm and 561-nm LD lasers (Yokogawa Electric) and an EMCCD camera (ImageEM 1K C9100-14 or ImageEM C9100-13; Hamamatsu Photonics, Shizuoka, Japan). Time-lapse images were acquired with a 40× objective lens (UPLSAPO40×, WD = 0.18 mm, NA = 0.95; Olympus). We used the 2 band-pass filters, 520/35 nm for the GFP and 617/73 nm for the tdTomato.

The number of observations and microscope information for each construct are listed in S3 Table.

## Isolation of female gametophyte cells

We used an inverted fluorescence microscope (IX-71; Olympus) equipped with a 3-charge-coupled device (CCD) digital camera (C7780; Hamamatsu Photonics). Images were acquired using a 40× objective lens (LUCPlanFl 40×, WD = 2.7–4 mm, NA = 0.60; Olympus). We emasculated the flowers at stage 12c [27]. Two days later, the unfertilized ovules of each cell marker line were treated with enzyme solution (1% cellulase [Worthington Biochemical Corporation, Freehold, USA], 0.3% macerozyme R-10 [Yakult, Tokyo, Japan], 0.05% pectolyase [Kyowa Kasei, Osaka, Japan], and 0.45 M mannitol [pH 7.0]). The target cells were collected during 20-minute to 1-hour period of the enzyme solution treatment; we used a micromanipulator (MN-4, MO-202U; Narishige, Tokyo, Japan) and micropipette (Picopipet HR; Nepa Gene, Chiba, Japan) with glass capillaries (G-1; Narishige), which were pulled with a micropipette

puller (P-97; Sutter, Novato, USA) [28]. The frequency of released central cells were lower than that of egg cells. After 30 minutes of treatment with the enzyme solution, ovules without released central cells were gently pressed by glass capillary to help release from the ovules.

### cDNA preparation and library construction for sequencing

The mRNA was extracted from 12 to 18 synergid, egg, and central cells with Dynabeads mRNA DIRECT Micro Kit according to the manufacturer's instructions (Invitrogen, Carlsbad, USA). The extracted mRNA was amplified using Ovation RNA sequencing (RNA-seq) System V2 (NuGEN, San Carlos, USA). The RNA-seq libraries were prepared using a TruSeq RNA Sample Preparation Kit and Multiplexing Sample Preparation Oligonucleotide Kit according to the manufacturer's instructions (Illumina, San Diego, USA). The libraries were sequenced on an Illumina GAIIx (Illumina) using 36-bp single-end reads.

### RNA-seq data analysis

Reads were filtered using fastp (ver. 0.20.0 [29]). The cleaned reads were mapped to the *Arabidopsis* reference genome TAIR10, using HISAT2 (ver. 2.1.0 [30]). The expression level for each gene was quantified using the read count and transcripts per million (TPM) with Stringtie (ver. 2.1.1 [31,32]). Differentially expressed genes (DEGs) between the synergid cells of the wild-type and the *myb98* mutant were identified by TCC with a false discovery rate of 0.01 (ver. 1.24.0 [33]). The TCC+baySeq (ver. 2.18.0) method with a false discovery rate of 0.01 was used for the identification of the DEGs among the synergid, egg, and central cells of the wild type [34]. Gene Ontology (GO) enrichment analysis was performed using g:Profiler (g:COSt; https://biit.cs.ut.ee/gprofiler/gost). The g:SCS threshold was set to 0.05.

## Results and discussion

### Live imaging of the nuclear dynamics during female gametophyte development

The development of female angiosperm gametophytes in vivo occurred within multiple layers of the maternal tissues of the flower. To investigate their actual developmental time course, we performed live-cell imaging of the female gametophytes development using the previously developed in vitro ovule culture system for embryogenesis in *Arabidopsis* [25]. To observe the nuclear dynamics in female gametophyte development, we constructed GPR1pro::H2B–mNeonGreen::GPR1ter (Fig 1A, S1 Movie). GPR1 (GTP-BINDING PROTEIN RELATED1) was previously found to be expressed in the megaspore mother cells (i.e., at stage FG0) and the female gametophytes at FG1 to FG7 [35]. At FG1, the nucleus was located at the female gametophyte center (Fig 1A; 0:00). Approximately 3 hours after the observation, the nucleus divided into two during its first mitosis (Fig 1B; 3:15). At FG2, the 2 nuclei were positioned at the center of the female gametophyte. Approximately 8 hours after the start of FG2, the nuclei moved to opposite ends of the ovule (Fig 1A; 12:00), at which point the vacuole may appear (FG3) [27]. After the second mitosis, the nuclei divide to lie in an orthogonal line along the chalazal–micropylar axis (Fig 1A; 13:00, 13:25). The chalazal nuclei migrated along a line that was parallel to the chalazal–micropylar axis (Fig 1A; 14:20), while the micropylar nuclei migrated along the surface of the female gametophyte, not parallel to the chalazal–micropylar axis (Fig 1A; 15:45, 18:40). The micropylar nuclei tended to lie along the abaxial surface of the female gametophytes (78/94, 83%). After the end of the third mitosis, the polar nuclei migrated linearly, not along the surface of the female gametophyte, but toward each other to fuse (Fig 1A; 20:40, 24:00). We calculated the duration of each nuclear division from 78 movies of *GPR1pro*::*H2B*–

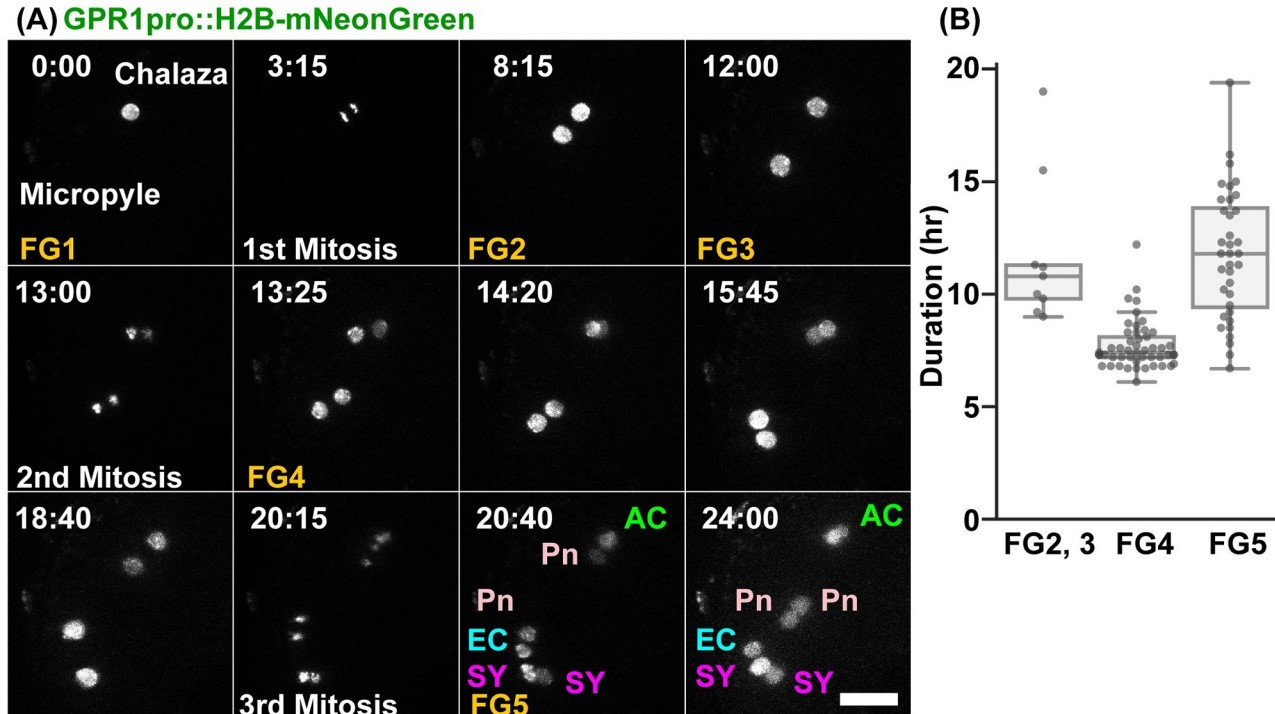

**Fig 1. Live-cell imaging of the nuclear dynamics during female gametophyte development in *Arabidopsis* using an in vitro ovule culture system.**
(**A**) Nuclei were labeled with *GPR1pro::H2B–mNeonGreen*. The numbers indicate time (hr:min) from the onset of observation. Images are representative of 37 time-lapse images from 3 independent transgenic lines. We succeeded in time-lapse recordings of the nuclear divisions in the isolated ovules from FG1 to FG6. FG1, uninucleate functional megaspore; FG2, 2-nucleate stage; FG3, 2 nuclei separated by a large central vacuole; FG4, 4-nucleate stage; FG5, 8-nucleate/7-celled stage; FG6, 7-celled with polar nuclei fused. Scale bar: 20 $\mu$m. (**B**) Durations of the nuclear divisions between the stages from FG2 to FG6. The interval times of the nuclear divisions for the female gametophyte development were analyzed for *GPR1pro::H2B–mNeonGreen* (*n* = 37), *RPS5Apro::H2B–tdTomato* (*n* = 29), and *RPS5Apro::H2B–sGFP* (*n* = 12). The underlying numerical data for B can be found in S1 Data. AC, antipodal cells; EC, egg cell; Pn, Polar nucleus; SY, synergid cell.

*mNeonGreen* (*n* = 37), *RPS5Apro::H2B–tdTomato* (*n* = 29), and *RPS5Apro::H2B–sGFP* (*n* = 12) (Fig 1B). The duration of the second and third nuclear divisions was 11.8 ± 3.3 hours (mean ± standard deviation; *n* = 9, Fig 1B; FG2,3) and 7.7 ± 1.1 hours (*n* = 49, Fig 1B; FG4), respectively. After cellularization, it took 3.8 ± 1.1 hours (*n* = 71) and 11.8 ± 2.9 hours (*n* = 36, Fig 1B; FG5) after the third mitosis for the polar nuclei to attach and fuse, respectively. Thus, normal female gametophyte development was observed using the in vitro ovule culture system [27]. The Nitsch medium supplemented with 5% trehalose resulted in the highest percentage of ovule survival in vitro during seed development and after fertilization [25]. This medium also enabled us to perform live-cell imaging during female gametophyte development, prior to fertilization.

## Live imaging of the plasma membrane formation during female gametophyte development

To analyze the relationship between the nuclear dynamics and plasma membrane formation during the cellularization, we observed their plasma membranes by labeling them with *RPS5A-pro::tdTomato–LTI6b* and nuclei with *RPS5Apro::H2B–sGFP* (Fig 2A, S2 Movie). The female gametophytes were located at the center of the ovule in the early stages of development (Fig 2A; −11:20). The female gametophytes showed polar elongation toward the micropylar ends of the ovule (Fig 2A; −6:00, −5:30, −4:45). The fluorescent signals of the *RPS5Apro::tdTomato–*

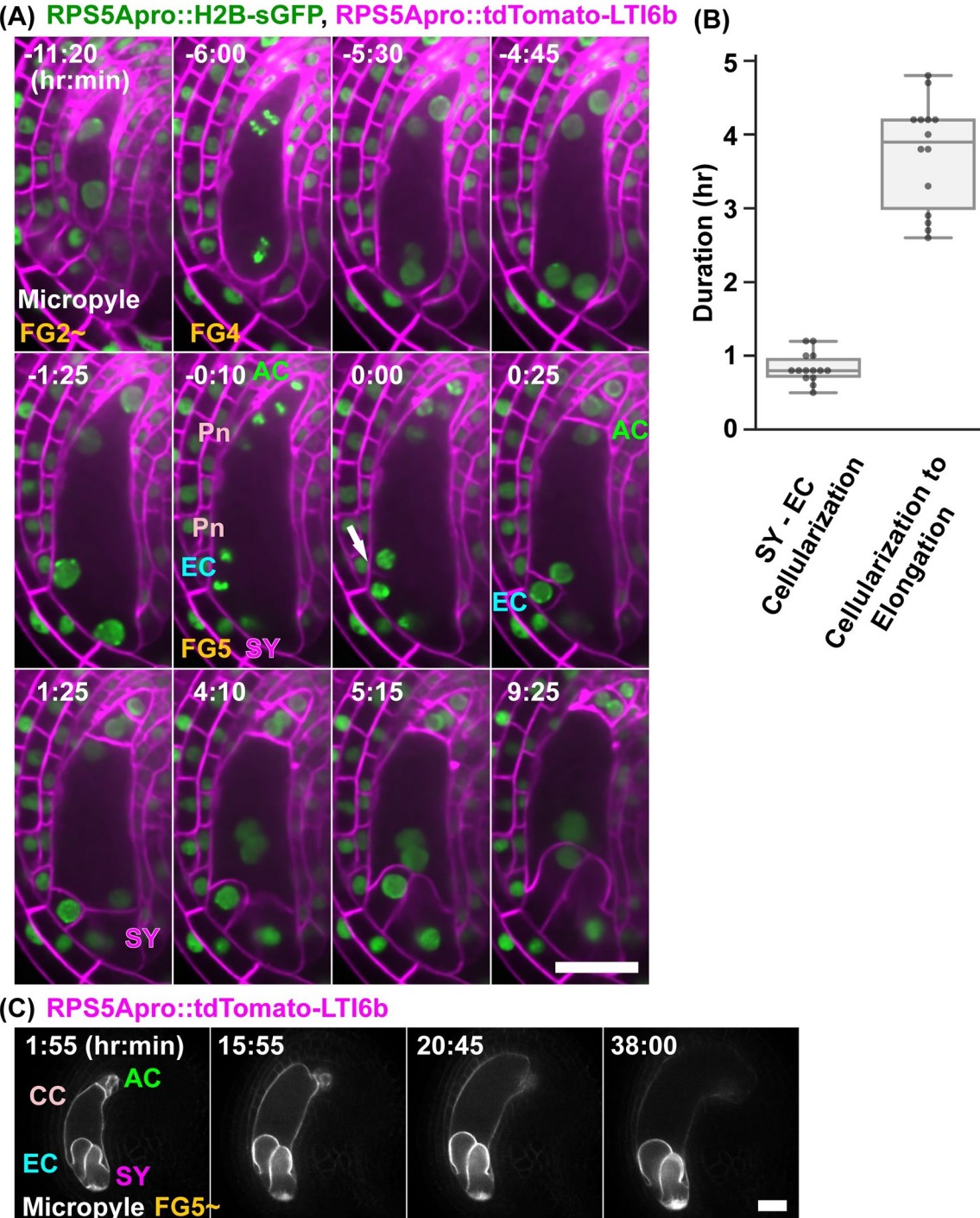

**Fig 2. Live-cell imaging of cellularization and maturation during FG4–FG5.** (**A**) Nuclei and plasma membranes were labeled with *RPS5Apro::H2B–sGFP* (green) and *RPS5Apro::tdTomato–LTI6b* (magenta), respectively. Numbers indicate time (hr:min) from the detection of the fluorescent signal of the tdTomato–LTI6b, on the forming cell plate (arrow). Images are representative of 10 time-lapse images from 4 independent transgenic lines. (**B**) Differences in the time to completion of the cellularization between the egg cell and synergid cells (left) and the initiation of the cell elongation from the completion of cellularization (right) at the FG5 stage. (**C**) Plasma membranes were labeled with *RPS5Apro::tdTomato–LTI6b*. Numbers indicate time (hr:min) from the onset of observation. Images are representative of 5 time-lapse images from a transgenic line. The underlying numerical data for B can be found in S1 Data. Scale bar: 20 μm. AC, antipodal cells; CC, central cell; EC, egg cell; Pn, Polar nucleus; SY, synergid cell.

*LTI6b* were detected in the plasma membranes of the female gametophytes during cellularization (Fig 2A; 0:00, arrow). Cellularization of the egg and synergid cells finished after 25 minutes and 1 hour and 25 minutes, respectively (Fig 2A). The time differences between the cellularization of the egg and the synergid cells was 0.8 ± 0.2 hours (*n* = 14; Fig 2B). After the cellularization, the egg and synergid cells were elongated toward the chalazal end (Fig 2A; 3:50, 7:35). It took 3.7 ± 0.7 hours (*n* = 14) from the completion of the cellularization to the start of the elongation (Fig 2B). In the case of the micropylar end, the fluorescent signals of the tdTomato–LTI6b were detected at the side nearest the nuclei, which gives rise to the polar nucleus and the egg nucleus after cellularization (Fig 2A; 0:00). This fluorescent signal was elongated to the opposite sides of the cell membranes of the female gametophytes. The polar nuclei migrated toward the opposite sides along with the plasma membrane formation (Fig 2A; 0:00 to 1:25). In the case of the chalazal end, the fluorescent signals of the tdTomato–LTI6b were also detected between the polar nucleus and the antipodal nucleus (S2 Movie). Thus, the dynamics of the plasma membrane formation were similar at the micropylar and chalazal ends.

During the maturation of the female gametophyte cells at the FG5 and FG6 stages, the central cell showed polar elongation toward the chalazal end of the ovule (Fig 2C, S3 Movie). A bright field movie showed that the central cell became elongated by collapsing the chalazal regions of the ovule (S3 Movie). The direction of this elongation was the opposite to that of the FG2 to FG4 (Fig 2A; −11:20 to −4:45). As shown in S3 Movie, the antipodal cells appeared to be collapsing during the maturation of the central cell. However, we could not determine whether the antipodal cells degenerated or not, i.e., whether they reached FG7 (4-celled stage) or not [36] in the *RPS5Apro::tdTomato–LTI6b*. Although we could not observe the signature of FG7, such as the degeneration of the antipodal cells, our in vitro culture system could monitor the entire development of the female gametophyte.

## Live imaging of cell fate specification during female gametophyte development

The transcriptome data of the mature ovules indicated that each female gametophyte cell had specific gene expression [37–39]. To investigate the initiation timing of the cell fate specification, we observed the egg cell marker, *EC1.2pro::mtKaede* [16], and the synergid cell marker, *MYB98pro::GFP–MYB98* [15] (Fig 3A and 3B, S4 and S5 Movies). The fluorescent signals of *EC1.2pro::mtKaede* were detected in the egg cells before their elongation (Fig 3A; 0:00). Considering that the duration from egg cell cellularization to egg cell elongation was about 4 hours (Fig 2B), *EC1.2* expression was initiated less than 4 hours after egg cell cellularization (Fig 2B). After 15 hours and 30 minutes, the fluorescent signals of *ABI4pro::H2B–tdTomato* were detected in the nucleus of the egg cell (Fig 3A; 21:10, arrowhead). Since MYB98 is an essential transcription factor for synergid cell function, the expression of MYB98 was predicted to begin after cellularization in the synergid cells; however, the fluorescent signals of *MYB98pro::GFP–MYB98* were detected in not only the synergid cells but also in the egg cell and the central cell around cellularization (Fig 3B; 0:05, 0:50). As the cells mature, the GFP signals were decreased in the egg and central cells but were maintained in the synergid cells (Fig 3B; 1:50, 3:10).

To determine when the expression of each cell-specific marker began after cellularization, we utilized the female gametophyte-specific markers *FGR8.0* [14] and *RPS5Apro::tdTomato–LTI6b* (Fig 3C, S6 Movie). Only the plasma membrane of female gametophyte cells was labeled, not sporophytic cells, in some RPS5Apro::tdTomato–LTI6b lines. After cellularization (Fig 3C; 0:00) and elongation of the egg and synergid cells (Fig 3C; 5:30), *EC1.1pro::NLS–3xDsRed2* and *LURE1.2pro::NLS–3xGFP* signals were detected in the egg and synergid cells, respectively, in

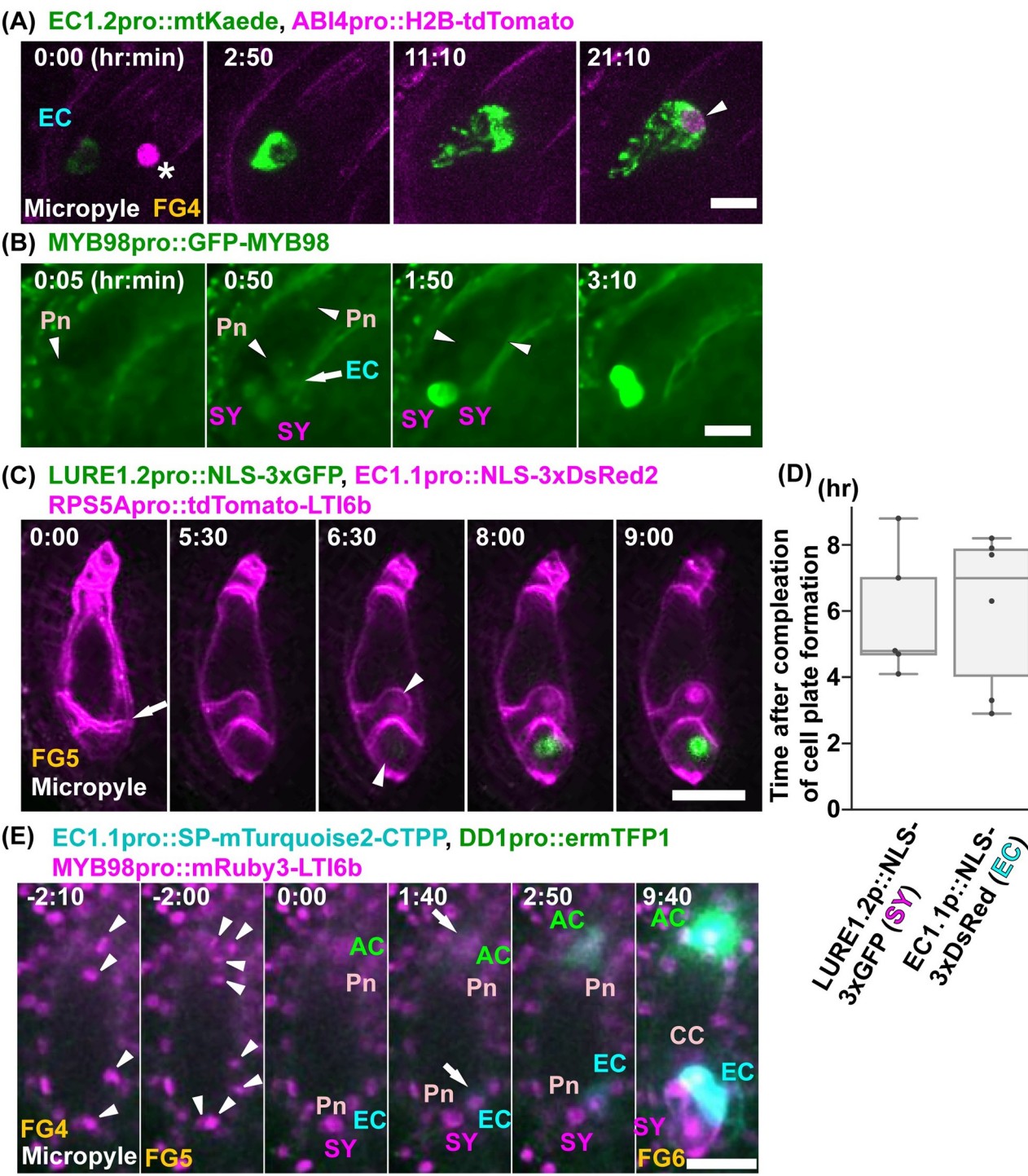

**Fig 3. Live-cell imaging of the cell fate specifications during FG4–FG5.** (**A**) The fluorescent signals of *EC1.2pro::mtKaede* were observed for the egg cell fate. Nuclei were labeled with *ABI4pro::H2B–tdTomato* (magenta). Numbers indicate time (hr:min) from the onset of observation. Images are representative of 4 time-lapse images from a transgenic line. Asterisk indicates the background signal in the ovule (0:00). Arrowhead indicates the fluorescent signal of *ABI4pro::H2B–tdTomato*. (**B**) The fluorescent signals of *MYB98pro::GFP–MYB98* were observed for the synergid cell fate in *myb98*. Numbers indicate the time (hr:min) after the detection of GFP signals. Images are representative of 23 time-lapse images from a transgenic line. (**C**) Nuclei were labeled with *EC1.1pro::NLS–3xDsRed2* (magenta) in the egg cells and *LURE1.2pro::NLS–3xGFP* (green) in the synergid cells, respectively, in *FGR8.0*. The plasma membranes were labeled with *RPS5Apro::tdTomato–LTI6b* (magenta). Numbers indicate the time (hr:min) after finishing the cell plate formation. Images are representative of 6 time-lapse images from a transgenic line. Arrow indicates the fluorescent signals of tdTomato–LTI6b on the forming cell plate. Arrowheads indicate the initiation of the expression of each cell-specific markers (6 hours 30 minutes). (**D**)

Initiation of the expression of the cell-specific markers at FG5. The fluorescent signals for *EC1.1pro::NLS–3xDsRed2* in the egg cells and *LURE1.2pro::NLS–3xsGFP* in the synergid cells were observed after completion of cell plate formation (**D**). (**E**) The fluorescent signals of *EC1.1pro::SP–mTurquoise2–CTPP*, *DD1pro::ermTFP1*, *MYB98pro::mRuby3–LTI6b*, and *AKVpro::H2B–mScarlet–I* were observed for the egg, antipodal, and synergid cell fates, and the nuclei movements, respectively. Numbers indicate time (hr:min) from the third mitosis. Images are representative of 2 time-lapse images from 2 independent transgenic lines. Arrowheads indicate the chromosomes during the third mitosis. Arrows indicate the initiation of the expression of the specific markers of the egg cell (cyan) and the antipodal cells (green) 1 hour and 40 minutes after cellularization. This timing was before the cell expansion and the polar nuclei migration. *MYB98pro::mRuby3–LTI6b* was detected 6 hours and 20 minutes after cellularization. The underlying numerical data for D can be found in S1 Data. Scale bars: 10 μm (**A**, **B**) and 20 μm (**C**, **E**). AC, antipodal cells; CC, central cell; EC, egg cell; Pn, Polar nucleus; SY, synergid cell.

*FGR8.0* (Fig 3C; 6:30). It took 5.9 ± 2.0 hours (*n* = 5) for the *EC1.1pro::NLS–3xDsRed2* to be detected after the completion of cellularization (Fig 3D). Considering that the expression of *EC1.2pro::mtKaede* was initiated before egg cell elongation (Fig 3A), the detection of the NLS marker was slower than that of the mitochondrial marker. *EC1.1* and *EC1.2* genes encode the cysteine-rich proteins and are specifically expressed in the egg cell in *Arabidopsis* [40]. It is possible that *EC1.1* (At1g76750) and *EC1.2* (At2g21740) are activated differently. Recently, Eason and colleagues developed a sensor for transiently expressed proteins (STEP) to detect protein expression rapidly in *Escherichia coli* [41]. The fluorescence of a dim GFP (gSTEP) increased 11-fold by binding to STEPtag within seconds. It would be useful to analyze the expression timing of gene/protein in vivo.

To investigate the correlation between the timing of the expression of each cell-specific marker at FG5, we used the multiple cell type–specific marker line (Fig 3E, S7 Movie). We changed the target signals of the new markers from the NLS and the fluorescent proteins as detection may have been slow. The cell-specific markers of the egg cell (*EC1.1pro::SP–mTurquoise2–CTPP*) and the antipodal cells (*DD1pro::ermTFP1*) were expressed 1 hour and 40 minutes after cellularization (Fig 3E; 1:40). This was before the egg and synergid cell elongations and the polar nuclei migrations. These results suggested that each cell fate was specified almost immediately after cellularization at the 8-nucleate stage.

## *myb98* synergid cells showed aberrant morphology and subcellular dynamics

MYB98 is required for the formation of the filiform apparatus during the synergid cell differentiation and the expression of the AtLURE1 peptides to attract the pollen tube in the synergid cells [10,42]. However, *MYB98pro::GFP–MYB98* was detected just after cellularization in the synergid, egg, and central cells (Fig 3B). To clarify the effects of the MYB98 transcription factor on the female gametophyte specifications, we observed the morphology and nuclear dynamics with the promoter activity of *MYB98* in the synergid cells of the wild-type and *myb98* mutant ovules (Fig 4, S8 and S9 Movies). The fluorescent signals of *MYB98pro::NLS–mRuby2* were also detected in the egg and central cells, as well as the synergid cells of the wild-type and *myb98* ovules. Although the nuclei were always located at the micropylar end of the synergid cells in the wild type (Fig 4A), they moved around in the synergid cells of *myb98* (Fig 4B). The nuclei tracking over 14 hours also showed that the nuclei of *myb98* moved closer to the chalazal end than in the wild type (Fig 4C). The large vacuoles occupied the chalazal end of the synergid cells in the wild type (Fig 4A). This polar distribution of the vacuole was disturbed in the synergid cells of *myb98* (Fig 4B). In addition, the *myb98* synergid cells were more elongated during maturation (Fig 4B; 2:50 to 8:20, S1 Fig). The results showed that the absence of *MYB98* affected the morphology and cellular dynamics of the synergid cells in addition to the formation of the filiform apparatus [10].

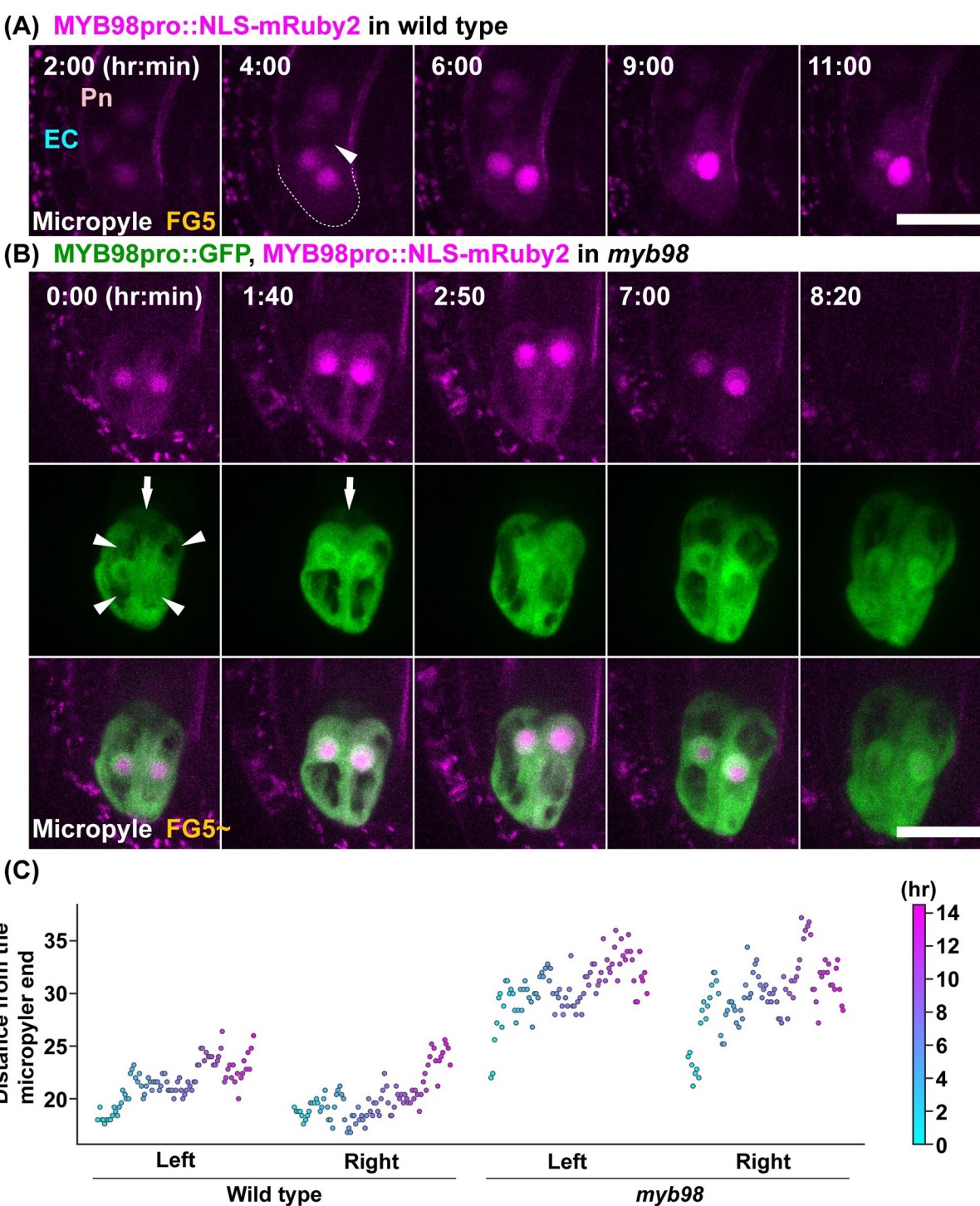

**Fig 4. Nuclear dynamics in the synergid cells of *myb98*.** (**A**) Nuclei of the synergid cells were labeled with *MYB98pro::NLS–mRuby2* in the wild type. The numbers indicate the time (hr:min) from the onset of observation. Images are representative of 6 time-lapse images from a transgenic line. Dashed lines indicate the surface of the synergid cells at the micropylar end. (**B**) Nuclei of the synergid cells that were labeled with *MYB98pro::NLS–mRuby2* in *myb98*. The fluorescent signals of *MYB98pro::GFP* were observed for the synergid cell fate. Images are representative of 12 time-lapse images from a transgenic line. The arrowheads indicate the vacuoles in the synergid cells. The arrows indicate the GFP signals in the egg cells. Scale bar: 20 *μ*m. (**C**) Nuclei positions on the micropylar–chalazal axis were plotted in each synergid cell in the wild type and *myb98* from S8 and S9 Movies. Each point indicates the time corresponding to the color bar. The leftmost point indicates the start time. The y-axis indicates the distance from the micropylar end of the synergid cell. The underlying numerical data for C can be found in S1 Data. EC, egg cell; Pn, Polar nucleus.

Previously, it has been reported that *MYB98pro::GFP* is expressed in all cells of the female gametophyte, except for the antipodal cells at FG5 [43]. The fluorescent signals of the GFP–MYB98 and NLS–mRuby2 were also detected in the synergid, egg, and central cells just after cellularization (Figs 3B and 4A). Except for the synergid cells, the fluorescent signals of GFP–MYB98 and NLS–mRuby2 were decreased as the cells matured. These results suggested that the synergid cell fate stabilized the gene expression of *MYB98*. The ectopic expression of *MYB98pro::GFP* and *MYB98pro::NLS–mRuby2* was not detected after the restrictions of the expression in the synergid cells of *myb98* mutant (Fig 4B). This suggested that the egg and central cells regularly maintain their cell fates. The promoter activity of *MYB98* was normal in *myb98* (Fig 4B), which also suggested that the maintenance, not initiation, of the synergid cell fate was defective in *myb98*. Considering these results, the positional information of the nuclei is essential for the initiation of the synergid cell fate.

## Gene expression analysis of the female gametophyte cell

Previous studies have supported the lateral inhibition model for the differentiation of the female gametophyte cells. Although all cells in the female gametophyte have the gametic cell competence, the accessory cells like the synergid and antipodal cells are repressed in the gametic cell fate [4,6]. To investigate the gene expression profiles of the synergid cells in the wild-type and *myb98* mutant, we established a method to isolate them in *Arabidopsis*. We treated the ovules in emasculated ovaries of the transgenic marker line for the synergid cells, *MYB98pro::GFP*, with enzyme solutions (Fig 5A). The protoplasts of the synergid cells were released from the ovules through their micropyles with enzyme treatment for 30 to 60 minutes (S2A Fig), and those with GFP signals were collected by micromanipulation (Fig 5A). Initially, the synergid cell–derived protoplasts were mostly associated with other GFP-negative ovular cells, probably due to insufficient cell wall digestion. To increase the efficiency of the single synergid cell isolation, we optimized the following 2 conditions. One was the calcium nitrate in the enzyme solution as the calcium ion was suggested to inhibit the degradation of the cell wall [44]. Subsequently, the removal of calcium ion from the enzyme solution decreased the adhesion of protoplasts and increased the frequency of the collectable synergid cells that were released as single cells (S2B Fig). The other condition was the pH of the enzyme solution. We found that the protoplasts began to decrease the GFP fluorescence in a short period and eventually ruptured after the cell surface gradually became rough, and this may be related to the decreases in viability. We performed the enzyme treatments at pH 5.0 to 9.0 and observed the GFP fluorescence as a vital indicator of the protoplast [45]. The rate of the GFP-positive synergid protoplasts was highest at pH 7.0, which was the best for the isolation of the synergid cells (Fig 2, S2 Fig). The optimized enzyme solutions allowed us to collect pure synergid cells with high efficiency (Fig 5C and 5D). To isolate other types of female gametophyte cells, we examined the enzyme solution treatment with the ovules of each marker line, *EC1.2pro::mtKaede* and *FWApro::FWA–GFP*, for the egg and central cells, respectively [16,17]. The protoplasts of the 2 gametic cells were also detached from their ovules through the micropyle (Fig 5F–5K).

We then performed RNA-seq to analyze the gene expression profiles of the collected the synergid, egg, and central cells in the wild type and the synergid cells in *myb98* mutant (Fig 5E). RNA-seq data from these female gametophyte cells were mapped to the genome of *Arabidopsis* (TAIR version 10) with the published sequence data from the ovules at 12 hour after emasculation (HAE) [46] and 2-week-old seedlings [47]. There were 5,007 to 13,073 genes (TPM > 1) detected in each sample (Fig 6A; S4 and S5 Tables). Transcripts from the same cell type in different biological replicates were highly correlated (S3A Fig). The principal component analysis (PCA) indicated that PC1 (31.8%) and PC2 (15.0%) were sufficient for separating

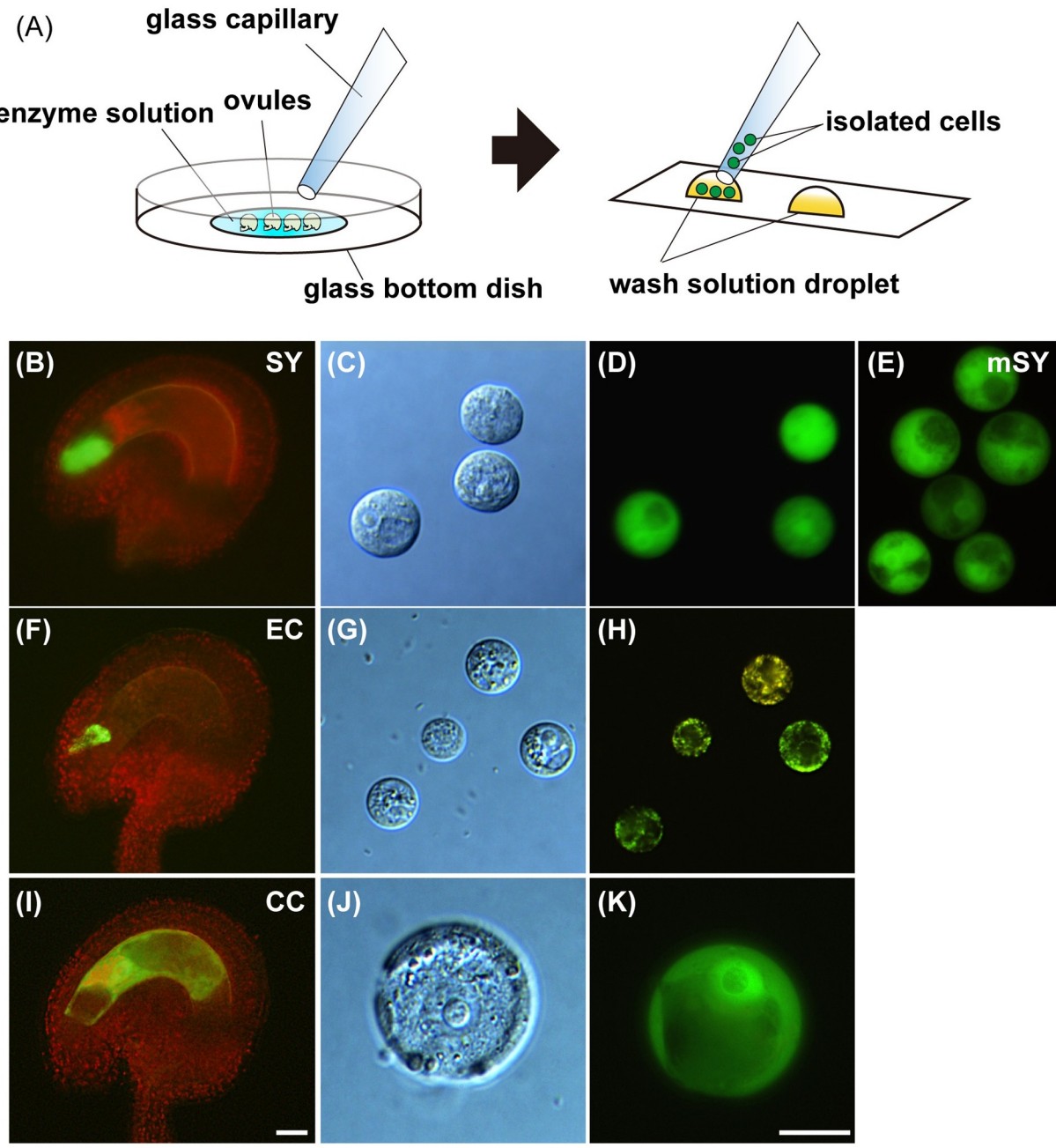

**Fig 5. Isolation of the female gametophyte cells.** (**A**) Scheme for the isolation of the female gametophyte cells. The ovules of the marker lines for the synergid (*MYB98pro::GFP*) (**B**), egg (*EC1.2pro::mtKaede*) (**F**), and central cells (*FWApro::FWA–GFP*) (**I**). We isolated each type of cell. Synergid cells in the wild-type (**C, D**) and *myb98* mutant (E). (**G, H**) Egg cells. (**J, K**) Central cell. Scale bar: 20 *μ*m. CC, central cell; EC, egg cell; mSY, synergid cell of *myb98* mutant; SY, synergid cell.

these samples into the 6 groups (Fig 6B, S3B and S3C Fig). These results suggested that our datasets had a high level of reproducibility and reflect the intermediary state of the *myb98* synergid. We identified the DEGs among the central, egg, and synergid cells in the wild type (S7–S9 Tables) and between the synergid cells in the wild-type and *myb98* mutant (S10 and S11 Tables). GO enrichment analysis of the DEGs that were specifically expressed in the wild-type female gametophyte cells indicated that the GO terms of each cell type were related to protein

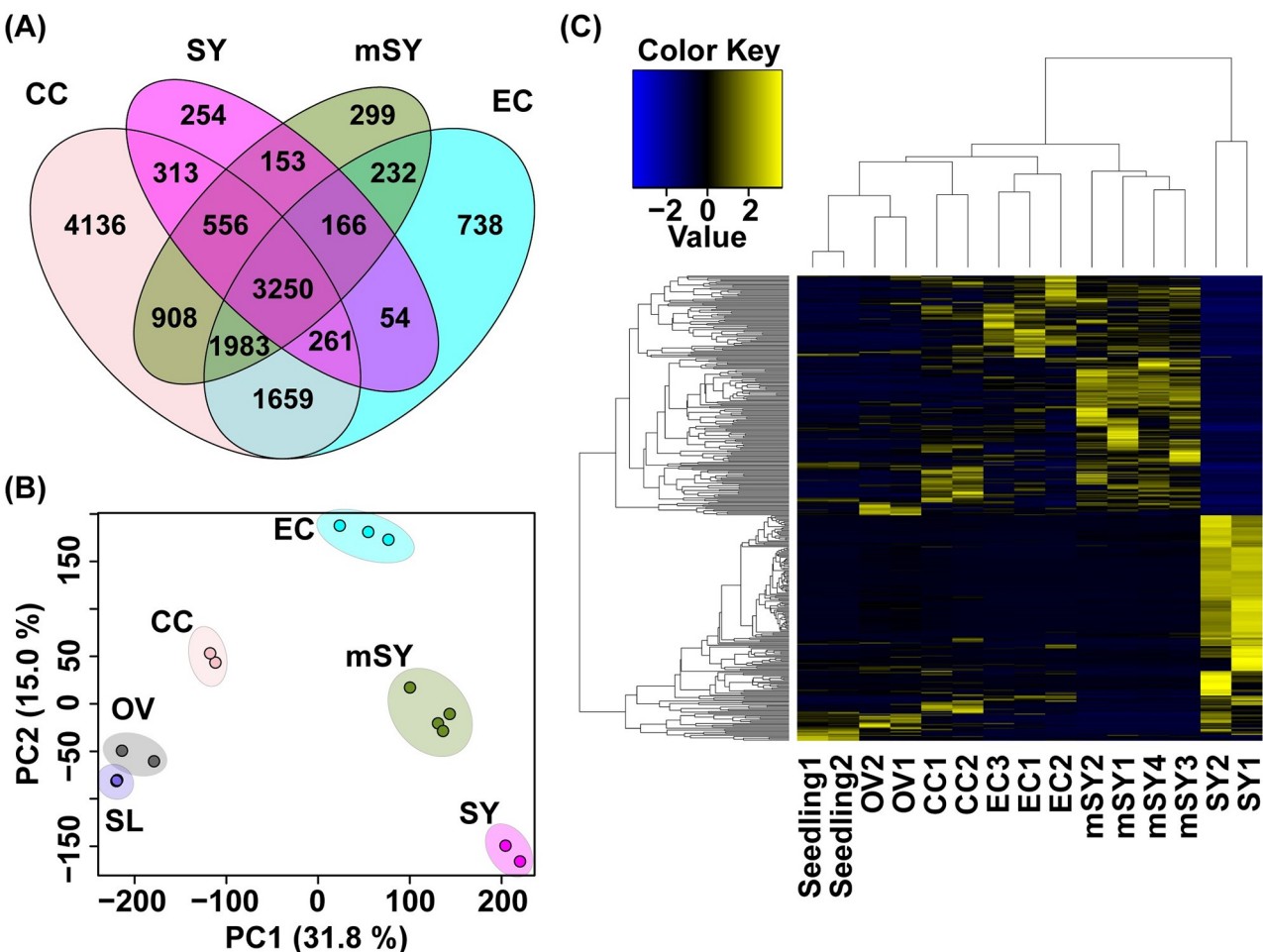

**Fig 6. RNA-seq of the female gametophyte cells.** The biological replicates were sequenced for 2 SY, 2 CC, 3 EC, and 4 mSY cells. (**A**) Venn diagram of the expressed genes (TPM > 1) in each cell type. (**B**) The PCA analysis (PC1 vs. PC2) of all transcriptome data, the female gametophyte cells, ovules, and seedlings. (**C**) Heatmap of the DEGs between the synergid cells in the wild-type and the *myb98* mutant. The underlying numerical data for (B, C) can be found in S1 Data. CC, central cell; EC, egg cell; mSY, synergid cell of *myb98* mutant; OV, ovule; RNA-seq, RNA sequencing; SL, seedling; SY, synergid cell.

secretion and cell wall synthesis in the synergid cells; fertilization, DNA methylation, and small regulatory RNA in the egg cells; and photosynthesis and interaction with other cells (cell killing and defense response) in the central cells (Tables 1 and 2, S12–S14 Tables). These results are consistent with the known functions of each cell type. The synergid cells play a role in pollen tube guidance through the secretion of many peptides, including *AtLUREs*, the filiform apparatus, and thickened cell wall structure [42]. GO terms supported that RNA-based gene silencing and DNA methylation mechanisms have important roles in the egg cells. The central cells play a role in pollen tube guidance through regulating the expression of cysteine-rich peptides (CRPs) from the synergid and central cells [42,48,49]. As in the present study, photosynthesis has recently been reported as a central cell–specific GO term [50]. Including other GO terms, the central cells have high metabolite levels (S14 Table). Interestingly, several egg cell–specific genes were highly expressed in the mutant synergid (S4 Fig, S6 Table). We examined the expression patterns of the DEGs in the synergid dataset among all samples (Fig 6C). The cluster of mutant synergids was closer to that of the egg cells than the synergid cells

**Table 1. GO enrichment analysis of DEGs in the synergid cells.**

| GO | Term ID | Term | *p*-value |
|---|---|---|---|
| MF | GO:0003978 | UDP-glucose 4-epimerase activity | 2.36E-02 |
| BP | GO:0051704 | multi-organism process | 3.25E-15 |
| BP | GO:0009567 | double fertilization forming a zygote and endosperm | 1.13E-14 |
| BP | GO:2000008 | regulation of protein localization to cell surface | 4.50E-14 |
| BP | GO:0034394 | protein localization to cell surface | 9.76E-14 |
| BP | GO:0080155 | regulation of double fertilization forming a zygote and endosperm | 1.60E-13 |
| BP | GO:1903827 | regulation of cellular protein localization | 9.37E-13 |
| BP | GO:0060341 | regulation of cellular localization | 3.79E-12 |
| BP | GO:0010183 | pollen tube guidance | 2.10E-06 |
| BP | GO:0005576 | extracellular region | 4.17E-09 |
| CC | GO:0031982 | vesicle | 2.45E-06 |
| CC | GO:0043680 | filiform apparatus | 6.55E-04 |
| CC | GO:0034663 | endoplasmic reticulum chaperone complex | 6.55E-04 |
| CC | GO:0005618 | cell wall | 1.03E-03 |

BP, biological process; CC, cellular component; DEG, differentially expressed gene; GO, Gene Ontology; MF, molecular function.

in the wild type. These results also indicated that the expression pattern of the *myb98* mutant synergid was partially changed to be egg cell–like.

Thus, the RNA-seq of the female gametophyte cells identified many of the DEGs and the highly expressed genes in each type of cell (S5–S11 Tables). We compared the DEGs between the wild type and *myb98* identified by this RNA-seq study with those identified by microarrays [38]. The number of up-regulated genes in *myb98* was 204 and 40 from the RNA-seq and microarray studies, respectively (S2D Fig). The number of down-regulated genes in *myb98* was 188 and 77 from the RNA-seq and microarray studies, respectively (S2E Fig). These results suggested that cell-specific RNA-seq had much higher sensitivity for the detection the DEGs than the microarrays because of the number of DEGs. Although 70 down-regulated genes in *myb98* overlapped between RNA-seq and microarray data, only 4 up-regulated genes in *myb98* overlapped (S2D and S2E Fig). The differences in the up-regulated genes of *myb98* may be caused by the wild type background or the developmental stage for the sampling [38]. Furthermore, our RNA-seq revealed that the gene expression profiles of the *myb98* mutant

**Table 2. GO enrichment analysis of DEGs in the egg cell.**

| GO | Term ID | Term | *p*-value |
|---|---|---|---|
| MF | GO:0061980 | regulatory RNA binding | 1.71E-03 |
| MF | GO:0016740 | transferase activity | 2.00E-02 |
| MF | GO:0010428 | methyl-CpNpG binding | 3.52E-02 |
| MF | GO:0010429 | methyl-CpNpN binding | 3.52E-02 |
| MF | GO:0070042 | rRNA (uridine-N3-)-methyltransferase activity | 3.52E-02 |
| BP | GO:0007338 | single fertilization | 1.40E-03 |
| BP | GO:0010629 | negative regulation of gene expression | 3.10E-02 |
| BP | GO:0051273 | beta-glucan metabolic process | 3.53E-02 |
| BP | GO:0006073 | cellular glucan metabolic process | 4.25E-02 |

BP, biological process; DEG, differentially expressed gene; GO, Gene Ontology; MF, molecular function.

synergid changed partially to the egg cell–like (S6 Table). The RNA-seq analysis conducted here allowed for the isolation of single cell types and mutants and thus enabled the detection of cell-specific changes. Our results provide evidence of the utility of this method for the investigation of cell fate specification mechanisms.

*MYB98* was reported as the gene that controlled the characteristic development of the synergid cells [10]. The *myb98* synergid was like a deficient egg cell because an important factor for the synergid cell fate was lost. The expression of AtLURE1 was also decreased in the *myb98* synergids [42]. In the present study, RKD1 and RKD2 and RKD2-induced genes were identified as egg cell DEGs, *myb98* DEGs [51]. RKD1 and RKD2 were up-regulated in a replicate of *myb98* synergid cells. RKD2-induced genes tended to be up-regulated in *myb98* synergid cells. Moreover, EC1.5 was detected as a DEG in *myb98* synergid cell. Other EC1s also tended to be up-regulated in *myb98* synergid cells. The PCA analysis and the difference of gene expressions reflect the intermediary state of *myb98* synergid (Fig 6B and 6C, S3B and S3C Fig). Further research is required to identify if the synergid cells of *myb98* function as egg cells, synergid cells, or both.

## Egg cell–specific markers were expressed in one of the synergid cells of *myb98*

To confirm the expression patterns of the egg cell–specific genes in *myb98*, we analyzed the CDR1–LIKE aspartyl proteases, which are highly expressed in the egg cells (S6 Table, S4 Fig). CONSTITUTIVE DISEASE RESISTANCE 1 (CDR1) was previously found to be involved in the peptide signaling of disease resistance [52]. The phylogenetic analysis showed that *Arabidopsis* contained 2 distinct groups of CDR1s: a CDR1–LIKE2 (At1g31450)/CDR1–LIKE1 group (At2g35615) and a CDR1 (At5g33340)/CDR1–LIKE3 (At1g64830) group [53] (S5 Fig). *CDR1–LIKE2pro*::*CDR1–LIKE2–mClover* (hereafter *CDR1L2–mClover*) and *CDR1–LIKE1pro*:: *CDR1–LIKE1–mClover* were expressed only in the egg cells, while *CDR1pro*::*CDR1–mClover* was expressed in the central and antipodal cells (S5B Fig, S10 Movie). These localizations were consistent with the groupings of the CDR1s by the phylogenetic analysis. Although the fluorescent signals of *CDR1L2–mClover* were limited to the egg cell after cellularization in the wild type (Fig 7A, S11 Movie, Table 3; 100%, *n* = 14), the *myb98* mutant had supernumerary cells with CDR1L2–mClover signals at the micropylar end (Fig 7B, S11 Movie, Table 3; 71%, *n* = 17). Initially, the CDR1L2–mClover signal was limited to a single cell at the egg cell position (Fig 7B; 0:00). However, 9 hours and 30 minutes after signal detection in the egg cell, the *CDR1L2–mClover* signal was also detected in one of the synergid cells (Fig 7B; 9:30). In most cases, one of the synergid cells expressed *CDR1L2–mClover* in *myb98* (Table 3; 65%, *n* = 17).

Previously, SBT4.13 was identified as an egg cell–specific gene [54]. We also analyzed the expression patterns of SBT4.13 in the wild-type and *myb98* mutant ovules (Fig 7C and 7D). The fluorescent signal of *SBT4.13pro*::*SBT4.13–mClover* was detected only in the egg cell before its elongation (Fig 7C; 0:00 to 1:30, S12 Movie). This expression timing of *SBT4.13pro*:: *SBT4.13–mClover* was similar to that of *EC1.2pro*::*mtKaede* (Fig 3A). The *myb98* ovules showed 2 patterns of *SBT4.13pro*::*SBT4.13–mClover* in the female gametophyte (Fig 7D, S13 Movie). One was the expression of *SBT4.13pro*::*SBT4.13–mClover* in the synergid and the antipodal cells in addition to the egg cells of the *myb98* ovules (S6A Fig; first half of S13 Movie; Table 3; 30%, *n* = 44). The other was the synergid and egg cells (Fig 7D; second half of S13 Movie, Table 3; 68%, *n* = 44). Similar to the results for *CDR1L2–mClover*, one of the synergid cells showed *SBT4.13pro*::*SBT4.13–mClover* expression in *myb98* (Table 3; 61%, *n* = 44).

To determine whether egg cell–specific markers were expressed in 1 or 2 synergid cells more clearly, we observed the *myb98* ovules in the multiple cell type–specific marker line (Fig 7E, S14 Movie). After detection of the *MYB98pro*::*mRuby3–LTI6b* signal in the 2 synergid

**Fig 7. Expression of EC–specific genes in the *myb98* synergid cells.** (**A**, **B**) The expression of *CDR1L2–mClover* in the wild-type (**A**) and *myb98* mutant (**B**) ovules. The numbers indicate the time (hr:min) from the onset of observation. Images are representative of 14 (**A**) and 17 (**B**) time-lapse images from 2 independent transgenic lines, respectively. The arrow indicates the CDR1L2–mClover signals in the synergid cell of *myb98*. (**C**, **D**) The expression patterns of *SBT4.13pro::SBT4.13–mClover* in the wild-type (**C**) and *myb98* mutant (**D**) ovules. The numbers indicate the time (hr:min) from the first detection of SBT4.13–mClover. Images are representative of 36 (**C**) and 44 (**D**) time-lapse images from 3 (**C**) and 4 (**D**) independent transgenic lines, respectively. The fluorescent signals of SBT4.13–mClover were only detected in the egg cells of the wild type. (**C**). However, in the case of *myb98*, the fluorescent signals

of SBT4.13–mClover were also detected in the synergid cells (**D**). (**E**) The expression patterns of the female gametophyte-specific markers in *myb98*. The numbers indicate the time (hr:min) from the onset of observation. At first, *MYB98pro::mRuby3–LTI6b* was detected in the 2 synergid cells (0:00). The arrows indicate the *EC.1pro::SP–mTurquoise2–CTPP* expression in one of the synergid cells (3 hours and 00 min, 5 hours and 00 min). The arrowhead indicates no expression of *EC.1pro::SP–mTurquoise2–CTPP* (3 hours and 00 min). The upper and lower panels indicate different *z* planes. Five ovules (50%, *n* = 10) showed EC1.1pro expression in the synergid cell, and the remaining 5 ovules were wild type–like. Scale bar: 20 *μ*m. AC, antipodal cells; EC, egg cell; SY, synergid cell.

cells, the signal of *EC.1pro::SP–mTurquoise2–CTPP* was detected in one of the synergid cells (Fig 7E; 3:00, 5:00, 8:00). Thus, one of the synergid cells showed cell fate conversion to an egg cell in *myb98*. In *myb98*, *SBT4.13pro::SBT4.13–mClover* was expressed almost simultaneously in the egg and synergid cells during cell elongation, whereas *CDR1L2–mClover* was expressed in the synergid cells after egg cell maturation. The expression of *SBT4.13pro::SBT4.13–mClover* in the synergid cells from an early stage indicated that *myb98* synergid cells had changed their cell fate from the early stage. These results suggest that the expression regulation of each egg cell–specific gene is different.

Although egg cell–specific genes were also expressed in one of the *myb98* synergid cells, the *myb98* pistils had only 1 embryo after fertilization, like the wild type (S7 Fig; 63 ovules from 10 pistils). This indicated that the synergid cells with the egg cell–specific genes were not functional for fertilization in *myb98*. The additional egg-like cells appeared to not be functional in *lis*, *clo*, *ato*, and *wyr* [6,8,9]. However, *amp1* has twin embryos, and *eostre* has twin zygote-like cells, indicating that these additional egg-like cells are functional for fertilization [5,7]. These differences in the gene expression of mutants may provide clues as to the acquisition of egg cell functions.

## Subcellular dynamics in female gametophyte development

To date, the female gametophyte of *Arabidopsis* has been analyzed only in fixed samples, so the actual developmental time course and subcellular dynamics were not known [27]. One of the major events that could not be seen in the fixed samples was that the vacuoles were dynamic in the female gametophytes. In the previous schematics, the vacuoles were drawn as large and only in the center of the cell [55]. When the polar nuclei migrated to fuse with each other at FG5, they were described as moving along the periphery of the female gametophyte to avoid the large vacuole [56]. However, the observations from the present study showed that the polar nuclei migrated linearly to fuse and adhere to the vacuole in the middle of the cell at shorter distances (Figs 1 and 8). This suggests that the vacuoles of the female gametophyte did not remain large and static, but changed shape dynamically. The dynamics of the vacuoles have been seen in *Arabidopsis* and tobacco BY-2 cultured cells, and this plasticity is due to actin filaments [57,58]. As actin filaments were also involved in the nuclear migrations during gamete

**Table 3. Expression of EC-specific genes in the female gametophyte cells.**

| Construct | MYB98 genotype | EC | EC AC | EC SY | EC SY AC | EC SY | EC SY AC |
|---|---|---|---|---|---|---|---|
| CDR1–LIKE2pro:: CDR1–LIKE2–mClover | +/+ | 14/14 (100%) | 0 | 0 | 0 | 0 | 0 |
| | −/− | 5/17 (29%) | 0 | 11/17 (65%) | 0 | 1/17 (6%) | 0 |
| SBT4.13pro:: BT4.13–mClover | +/+ | 34/36 (94%) | 2/36 (6%) | 0 | 0 | 0 | 0 |
| | −/− | 1/44 (2%) | 0 | 27/44 (61%) | 11/44 (25%) | 3/44 (7%) | 2/44 (5%) |

AC, antipodal cells; EC, egg cell; SY, synergid cell.

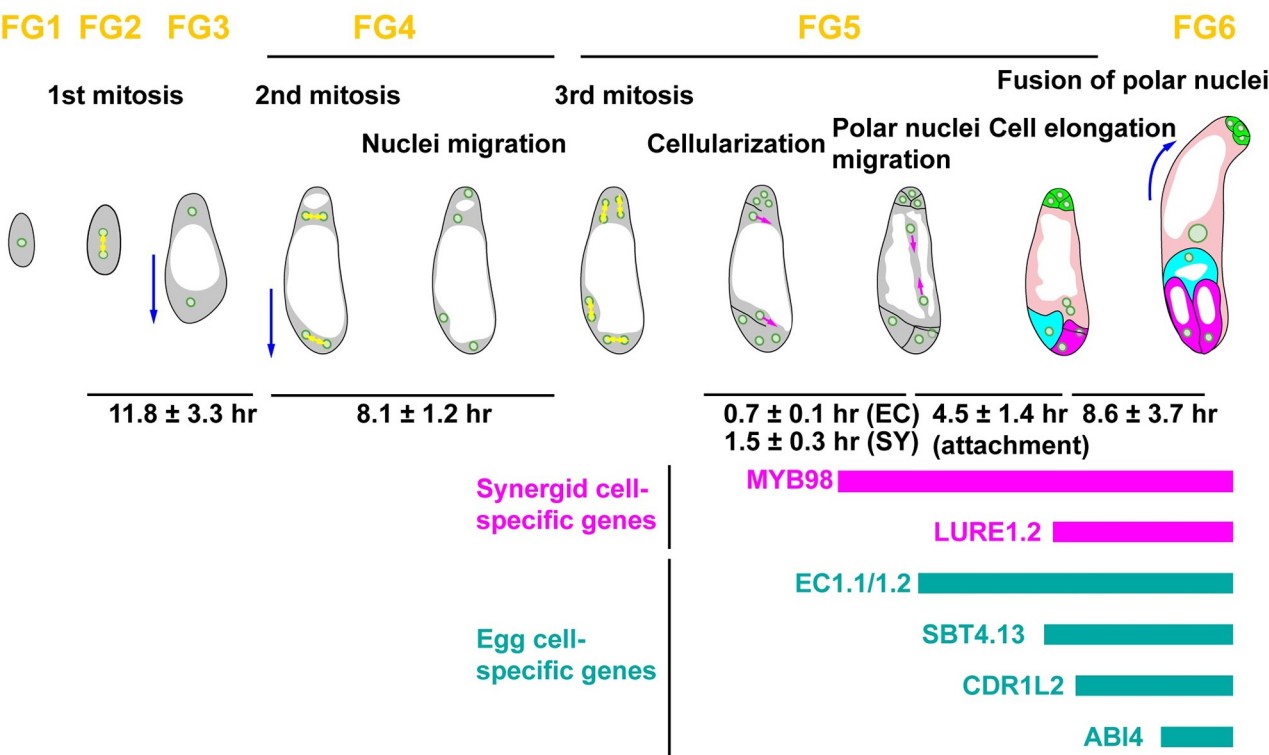

**Fig 8. Schematic illustration of the dynamics of female gametophyte development in *Arabidopsis*.** Yellow arrows show the direction of nuclear divisions. Blue arrows show the direction of cell elongation of the female gametophyte. Magenta arrows show polar nuclear migration at FG5. The time (mean ± standard deviation) was calculated from the movies. EC, egg cell; SY, synergid cell.

fusion, the linear migration of the polar nuclei was also expected to involve them [20]. In the mature central cells after polar nuclei fusions, the nucleus of the central cells was located to the micropylar end, and the actin filaments played an important role in the positioning of the nucleus [59]. The vacuoles were located at the chalazal end of the synergid cells and the micropylar end of the egg cells, thus appearing to limit the nuclear migration (Fig 3A and 3B). In the case of the *myb98* mutant, the vacuoles were dynamic, causing the nuclei to move around and not stay in one place (Fig 4B and 4C). It is considered that this nuclear movement promoted the expression of the egg cell markers in the synergid cells of *myb98*. The gene expression analysis showed an intermediary state between the synergid and egg cells in the *myb98* synergid, suggesting that this nuclear movement may appear as a mixture of that seen in egg and synergid cells (Fig 6B and 6D, S3B and S3C Fig). Strong correlations between the nuclear position and the cell fate were shown in several mutants [5–9]. However, it remains unclear whether the nuclear position determines gene expression or if gene expression determines the nuclear position. Manipulation of nuclear behavior with the in vitro ovule culture systems will help to reveal the mechanisms of cell fate specifications in the development of the female gametophytes.

## Cell–cell communication between the 2 synergid cells

An interesting phenotype of the *myb98* mutant was in one of the 2 synergid cells that tended to be converted to an egg cell fate (Table 3; 92% for *CDR1L2–mClover*, n = 12, 88% for *SBT4.13pro::SBT4.13–mClover*, n = 43). Some mutants show similar phenotypes with additional egg cells [5–9]. In the *amp1* mutant, 19% of the ovules showed *EC1.1pro::HTA6–3GFP*

expression in both synergid cells, whereas 26% of the ovules showed this expression in only one of the synergid cells (notably, 45% of the ovules had no detectable fluorescent signal) [5]. The combination of cell–cell communication and flexible fate maintenance might allow for only one of the 2 synergid cells to become an egg cell. The synergid cells play an important role in pollen tube attraction through secreting peptides [60]. In the case of the ovule, which has been converted from both synergid cells to the egg cell fate, it cannot attract pollen tubes. Therefore, it is expected that plants may have a mechanism, which is independent of the MYB98, to retain not only the egg cell but also the synergid cell for pollen tube attraction and fertilization. Previously, we found that the laser disruption of the immature egg cells affects the cell differentiation for one of the synergid cells in *Torenia fournieri* [61]. Lateral inhibition from the egg cell restricts the egg cell fate in accessory cells [14]. Based on these findings, we speculate that egg cell failure induced a decrease in *MYB98* expression. The results presented here raise 2 possibilities. One possibility is that the synergid cell acquired the egg cell fate, preventing the remaining synergid cell from obtaining the egg cell fate. However, we observed egg cell–specific gene expression in both synergid cells in only a few cases (11%, $n = 55$). In this case, the inhibition signal needs to be quickly transmitted to the remaining synergid cell. The other possibility is that the cell–cell communication between the 2 synergid cells decides which cells convert to the egg cell fate. Exploration and imaging of signal molecules involved in this cell–cell communication would help to reveal which of these possibilities occurs.

Our results suggested that the cell fate specifications are immediately initiated around the time of cellularization, depending on the positional information of the nucleus. Moreover, failure of cell fate maintenance, like that of the *myb98* mutant, induced cell fate conversion from the adjacent accessory cells to gamete cell fates for fertilization. Previously, the existence of cell–cell communication between gametic and accessory cells, such as lateral inhibition from the egg cell to synergid cells, was proposed [4]. We propose that the synergid cells communicate with each other to determine their fate and behavior, and such flexibility compliments for the robustness of plant fertilization. Further studies, such as single-cell transcriptome profiling of the mutant synergids, will provide novel insights into the molecular mechanisms of the cell–cell communications in the cell fate specification of plants.

## Supporting information

**S1 Fig. Cell length during synergid cell maturation.** We measured the longitudinal length of the synergid cells every 1 hour for *MYB98pro::NLS–mRuby2* in a wild-type ovule and *MYB98-pro::NLS–mRuby2* and *MYB98pro::GFP* in a *myb98* ovule from S8 and S9 Movies. The underlying numerical data for this figure can be found in S1 Data.
(TIF)

**S2 Fig. RNA-seq of the female gametophyte cells. (A)** Two synergid cells were released from the ovules of *MYB98pro::GFP*. **(B)** Frequency of the collectable synergid cells with or without calcium nitrate in the enzyme solution. An F-test of the frequency showed that the variances of the groups with and without calcium nitrate were equal ($p > 0.05$). The results of Student $t$ test showed that the absence of calcium nitrate was more effective for synergid cell isolation (${}^{*}p < 0.05$). **(C)** Frequency of the collectable synergid cells depending on the pH of the enzyme solution. An F-test of the frequency showed that the variances between pH 6 and pH 7 and pH 7 and pH 8 were equal ($p > 0.05$). A result of Student $t$ test showed that the frequency of collectable synergid cells was different significantly between pH 6 and pH 7 and pH 7 and pH 8 (${}^{*}p < 0.05$; ${}^{**}p < 0.01$). **(C, D)** Venn diagram of DEGs that were up-regulated **(D)** or down-regulated **(E)** in the *myb98* mutant synergids between the RNA-seq and microarray. The underlying numerical data for (B, C) can be found in S1 Data. DEG, differentially expressed

gene; RNA-seq, RNA sequencing.
(TIF)

**S3 Fig. RNA-seq analysis of female gametophyte cells. (A)** The Pearson correlation of RNA-seq libraries. **(B, C)** The PCA analysis (PC1 vs. PC3, PC2 vs. PC3) of all transcriptome data. The underlying numerical data for this figure can be found in S1 Data. PCA, principal component analysis; RNA-seq, RNA sequencing.
(TIF)

**S4 Fig. The gene expression of highly expressed 50 DEGs in egg cells and central cells.** Highly expressed genes of *myb98* synergid cells were more abundant in DEGs of egg cells than those of central cells. The underlying numerical data for this figure can be found in S1 Data. DEG, differentially expressed gene.
(TIF)

**S5 Fig. The expression patterns of CDR1s in the female gametophyte. (A)** Phylogenetic tree of the aspartyl proteases in the *Arabidopsis thaliana*. **(B)** The expression patterns of the *CDR1L2–mClover* and *CDR1L1–mClover* were detected in the egg cell. The fluorescent signal of the *CDR1–mClover* was detected in the central cell and the antipodal cells. Scale bar: 20 $\mu$m.
(TIF)

**S6 Fig. The expression patterns of *SBT4.13pro::SBT4.13–mClover* in the *myb98* mutant ovules.** The numbers indicate the time (hr:min) from the first detection of the SBT4.13–mClover. The fluorescent signals of the SBT4.13–mClover were also detected in the synergid cells and the antipodal cells. Scale bar: 20 $\mu$m.
(TIF)

**S7 Fig. The embryo development in *myb98* mutant ovules.** DIC images showed the cleared globular embryos of the *myb98* mutant. Scale bar: 20 $\mu$m. DIC, disseminated intravascular coagulation.
(TIF)

**S1 Movie. Nuclear dynamics during female gametophyte development in *Arabidopsis thaliana*.** Time-lapse movie of a *GPR1pro::H2B–mNeonGreen* ovule. Images were taken at 5-minute intervals, and the movie is displayed at 30 frames per second. Scale bar: 20 $\mu$m (see also Fig 1A).
(MOV)

**S2 Movie. Morphological changes and plasma membrane formation during female gametophyte development.** Time-lapse movie of an *RPS5Apro::tdTomato–LTI6b* ovule. Images were taken at 5-minute intervals, and the movie is displayed at 30 frames per second. Scale bar: 20 $\mu$m (see also Fig 2A).
(MOV)

**S3 Movie. The maturation of female gametophyte cells after cellularization.** Time-lapse movies of *RPS5Apro::tdTomato–LTI6b* ovules. Images were taken at 5-minute intervals, and the movie is displayed at 30 frames per second. Scale bar: 20 $\mu$m (see also Fig 2C).
(MOV)

**S4 Movie. Expression of egg cell–specific markers at FG5.** Time-lapse movie of a *EC1.2pro::mtKaede* (green) and *ABI4pro::H2B–tdTomato* (magenta) ovule. Images were taken at 10-minute intervals, and the movie is displayed at 15 frames per second. Scale bar: 20 $\mu$m (see also Fig 3A).
(MOV)

**S5 Movie. Expression of synergid cell–specific marker at FG4 and FG5.** Time-lapse movie of a *MYB98pro*::*GFP–MYB98* (green) and *RPS5Apro*::*H2B–tdTomato* (magenta) ovule. Images were taken at 5-minute intervals, and the movie is displayed at 15 frames per second. Scale bar: 20 *µ*m (see also Fig 3B).
(MOV)

**S6 Movie. Expression of egg cell and synergid cell–specific markers after cellularization.** Time-lapse movie of an *RPS5Apro*::*tdTomato–LTI6b* (magenta), *EC1.1pro*::*NLS–3xDsRed* (magenta), and *LURE1.2pro*::*NLS–3xGFP* (green) ovule. Images were taken at 5-minute intervals, and the movie is displayed at 30 frames per second. Scale bar: 20 *µ*m (see also Fig 3C).
(MOV)

**S7 Movie. Expression of egg cell, synergid cell, and antipodal cell–specific markers after cellularization.** Time-lapse movie of a *EC1.1pro*::*SP-mTurquoise2–CTPP* (cyan), *MYB98pro*::*mRuby3–LTI6b* (magenta), *DD1pro*::*ermTFP1* (green), and *AKVpro*::*H2B–mScarlet-I* (magenta) ovule. Images were taken at 10-minute intervals, and the movie is displayed at 15 frames per second. Scale bar: 20 *µ*m (see also Fig 3E).
(MOV)

**S8 Movie. Expression of *MYB98pro*::*NLS-mRuby2* in the wild type.** Time-lapse movie of *MYB98pro*::*NLS–mRuby2* in a wild-type ovule. Images were taken at 10-minute intervals, and the movie is displayed at 15 frames per second. Scale bar: 20 *µ*m (see also Fig 4A).
(MOV)

**S9 Movie. Expression of *MYB98pro*::*NLS-mRuby2* in *myb98*.** Time-lapse movie of *MYB98-pro*::*NLS–mRuby2* (magenta) and *MYB98pro*::*GFP* (green) in a *myb98* ovule. Images were taken at 10-minute intervals, and the movie is displayed at 15 frames per second. Scale bar: 20 *µ*m (see also Fig 4B).
(MOV)

**S10 Movie. Expressions of CDR1s–mClover after cellularization.** Time-lapse movies of a *CDR1–LIKE2pro*::*CDR1–LIKE2–mClover* (green) ovule in the first movie, a *CDR1–LIKE1pro*::*CDR1–LIKE1–mClover* (green) ovule in the second movie, and a *CDR1pro*::*CDR1–mClover* (green) ovule in the third movie. Images were taken at 15-minute intervals, and the movies are displayed at 15 frames per second. Scale bar: 20 *µ*m (see also S4 Fig).
(MOV)

**S11 Movie. Expression of *CDR1–LIKE2pro*::*CDR1–LIKE2–mClover* in *myb98*.** Time-lapse movie of *CDR1–LIKE2pro*::*CDR1–LIKE2–mClover* in a *myb98* ovule. Images were taken at 10-minute intervals, and the movie is displayed at 15 frames per second. Scale bar: 20 *µ*m (see also Fig 7B).
(MOV)

**S12 Movie. Expression of *SBT4.13pro*::*SBT4.13–Clover* in the wild type.** Time-lapse movie of *SBT4.13pro*::*SBT4.13–mClover* in a wild-type ovule. Images were taken at 10-minute intervals, and the movie is displayed at 15 frames per second. Scale bar: 20 *µ*m (see also Fig 7C).
(MOV)

**S13 Movie. Expression of *SBT4.13pro*::*SBT4.13–Clover* in *myb98*.** Time-lapse movie of *SBT4.13pro*::*SBT4.13–Clover* in a *myb98* ovule. Images were taken at 10-minute intervals, and the movie is displayed at 15 frames per second. Scale bar: 20 *µ*m (see also Fig 7D).
(MOV)

**S14 Movie. Expression of egg cell, synergid cell, and antipodal cell–specific markers in
*myb98*.** Time-lapse movie of *EC1.1pro::SP-mTurquoise2–CTPP* (cyan), *MYB98pro::mRuby3–
LTI6b* (magenta), and *DD1pro::ermTFP1* (green) in a *myb98 ovule*. Images were taken at
10-minute intervals, and the movie is displayed at 15 frames per second. Scale bar: 20 *µ*m (see
also Fig 7E).
(MOV)

**S1 Data. Numerical raw data in Figs 1B, 2B, 3D, 4C, 6B and 6D and S1, S2B, S2C, S3 and
S4 Figs.**
(XLSX)

**S1 Table. Transgenic lines used in this study.**
(XLSX)

**S2 Table. Primers used in this study.**
(XLSX)

**S3 Table. The number of observations and microscope information for each construct.**
(XLSX)

**S4 Table. RNA-seq libraries of female gametophyte cells.** Total reads and mapping rates to
the genome of *Arabidopsis thaliana* for each RNA-seq library. RNA-seq, RNA sequencing.
(XLSX)

**S5 Table. TPM of expressed genes and statistics for DEGs identification.** TPM value of
the genes (TPM > 1) and statistical processing for DEGs identification among egg, cenrtral
and synergid cells or between synergid cells in wild-type and *myb98* mutant. The numbers
in the row of nonDEG, DEG_CC, DEG_EC, DEG_SY, and DEG_all were the probability for
the each DEGs on the comparison with female gametophyte in wild type. A.value and M.
value mean log-intensity average and log-intensity ratios of the expression level in the
wild-type and *myb98* synergid cells. DEG, differentially expressed gene; TPM, transcripts
per million.
(XLSX)

**S6 Table. Egg specific genes and DEGs between wild-type and *myb98* synergid cells.** The
genes specifically expressed in the egg cells that were DEGs between wild-type and *myb98* syn-
ergid cells. DEG, differentially expressed gene.
(XLSX)

**S7 Table. DEGs in egg cell among wild-type female gametophyte cells.** The DEGs in the egg
cells among female gametophyte cells. DEG, differentially expressed gene.
(XLSX)

**S8 Table. DEGs in central cell among wild-type female gametophyte cells.** The DEGs in the
central cells among female gametophyte cells. DEG, differentially expressed gene.
(XLSX)

**S9 Table. DEGs in synergid cell among wild-type female gametophyte cells.** The DEGs in
the synergid cells among female gametophyte cells. DEG, differentially expressed gene.
(XLSX)

**S10 Table. Up-regulated genes in *myb98* synergid cells compared to those of wild type.**
Highly expressed genes in *myb98* synergid cells.
(XLSX)

**S11 Table. Down-regulated genes in *myb98* synergid cells compared to those of wild type.** Slightly expressed genes in *myb98* synergid cells.
(XLSX)

**S12 Table. GO enrichment analysis of the DEGs which expressed in the synergid cells among female gametophyte cells of wild type.** BP, biological process; CC, cellular component; DEG, differentially expressed gene; GO, Gene Ontology; MF, molecular function.
(XLSX)

**S13 Table. GO enrichment analysis of the DEGs which expressed in the egg cells among female gametophyte cells of wild type.** DEG, differentially expressed gene; GO, Gene Ontology.
(XLSX)

**S14 Table. GO enrichment analysis of the DEGs which expressed in the central cells among female gametophyte cells of wild type.** DEG, differentially expressed gene; GO, Gene Ontology.
(XLSX)

## Acknowledgments

We thank R. Groß-Hardt, G. Drews, T. Kinoshita, M. Fujimoto, and R. D. Kasahara for the plant materials; S. Nasu, T. Nishii, T. Shinagawa, and Y. Taniuchi for assistance with cloning and the generation of transgenic plants; T. Nagata, N. Kurata, J. Kawarama, H. Ohyanagi, A. Toyoda, and A. Fujiyama for support to examine the mRNA method of amplification; H. Nagata and K. Tonosaki for discussions regarding the data analysis; and Editage (www.editage.com) for English language editing. Computations were partially performed on the NIG supercomputer at ROIS National Institute of Genetics.

## Author Contributions

**Conceptualization:** Daichi Susaki, Daisuke Kurihara.

**Data curation:** Daichi Susaki, Takamasa Suzuki.

**Formal analysis:** Daichi Susaki, Takamasa Suzuki.

**Funding acquisition:** Daichi Susaki, Daisuke Maruyama, Minako Ueda, Tetsuya Higashiyama, Daisuke Kurihara.

**Investigation:** Daichi Susaki, Takamasa Suzuki, Daisuke Maruyama, Minako Ueda, Tetsuya Higashiyama, Daisuke Kurihara.

**Methodology:** Daichi Susaki, Takamasa Suzuki, Daisuke Kurihara.

**Project administration:** Daichi Susaki, Tetsuya Higashiyama, Daisuke Kurihara.

**Resources:** Daichi Susaki, Takamasa Suzuki, Daisuke Maruyama, Tetsuya Higashiyama, Daisuke Kurihara.

**Software:** Takamasa Suzuki.

**Supervision:** Daichi Susaki, Tetsuya Higashiyama, Daisuke Kurihara.

**Validation:** Daichi Susaki, Daisuke Kurihara.

**Visualization:** Daichi Susaki, Daisuke Kurihara.

**Writing – original draft:** Daichi Susaki, Daisuke Kurihara.

**Writing – review & editing:** Takamasa Suzuki, Daisuke Maruyama, Minako Ueda, Tetsuya Higashiyama.

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
