## [Editor Report · Decision Letter 0]

29 Jul 2020

Dear Dr Kurihara, 

Thank you for submitting your manuscript entitled "Dynamics of the cell fate specifications during female gametophyte development in Arabidopsis" for consideration as a Research Article by PLOS Biology. Thank you also for your patience as we completed our editorial process, and please accept my apologies for the delay in providing you with our decision.

Your manuscript has now been evaluated by the PLOS Biology editorial staff as well as by an academic editor with relevant expertise and I am writing to let you know that we would like to pursue the manuscript for publication. In order to do so, we need you to complete your submission by providing the metadata that we required. To this end, please login to Editorial Manager where you will find the paper in the 'Submissions Needing Revisions' folder on your homepage. Please click 'Revise Submission' from the Action Links and complete all additional questions in the submission questionnaire.

Once we have this, we will send you a decision letter inviting you to revise the manuscript along the lines you indicate in your rebuttal.

Please re-submit your manuscript within two working days, i.e. by Jul 31 2020 11:59PM.

Kind regards,

Ines

--

Ines Alvarez-Garcia, PhD

Senior Editor

PLOS Biology

---

## [Editor Report · Decision Letter 1]

31 Jul 2020

Dear Dr Kurihara,

Thank you very much for submitting your manuscript "Dynamics of the cell fate specifications during female gametophyte development in Arabidopsis" for consideration as a Research Article at PLOS Biology. As you know, your manuscript and plan of revision have been evaluated by the PLOS Biology editors and by an Academic Editor with relevant expertise.

Based on your responses to the reviews from Reviews Commons, we would welcome re-submission of a revised version that takes into account the reviewers' comments. We cannot make any decision about publication until we have seen the revised manuscript and your response to the reviewers' comments. Your revised manuscript is also likely to be sent for further evaluation by the original reviewers.

We expect to receive your revised manuscript within 3 months. 

**IMPORTANT - SUBMITTING YOUR REVISION**

*Re-submission Checklist*

*Published Peer Review*

*PLOS Data Policy*

*Blot and Gel Data Policy*

Sincerely,

Ines

--

Ines Alvarez-Garcia, PhD,

Senior Editor,

ialvarez-garcia@plos.org,

PLOS Biology

---

## [Decision Letter · Decision Letter 2]

23 Dec 2020

Dear Dr Kurihara,

Thank you for submitting your revised Research Article entitled "Dynamics of the cell fate specifications during female gametophyte development in Arabidopsis" for publication in PLOS Biology. I have now obtained advice from two of the original reviewers and have discussed their comments with the Academic Editor. 

Based on the reviews, we will probably accept this manuscript for publication, assuming that you will modify the manuscript to address the remaining points made by Reviewer 1 and the data and other policy-related requests noted at the end of this email.

We expect to receive your revised manuscript within two weeks.

-  a cover letter that should detail your responses to any editorial requests.

*Published Peer Review History*

*Early Version*

Sincerely,

Ines

--

Ines Alvarez-Garcia, PhD,

Senior Editor,

ialvarez-garcia@plos.org,

PLOS Biology

ETHICS STATEMENT:

-- Thank you for providing an ethic statement in the Methods section. Please include the protocol/permit/project license.

Fig. 1I; Fig. 2B, F; Fig. 3A, D, F, I, K; Fig. 4A, D, F; Fig. 5A-D, G, I, K; Fig. S1D and Fig. S3B, D

Please also ensure that figure legends in your manuscript include information on WHERE THE UNDERLYING DATA CAN BE FOUND.

Please also make publicly available all the data you have deposited in SRA, otherwise we won't be able to proceed with production.

BLURB

Please also provide a blurb which (if accepted) will be included in our weekly and monthly Electronic Table of Contents, sent out to readers of PLOS Biology, and may be used to promote your article in social media. The blurb should be about 30-40 words long and is subject to editorial changes. It should, without exaggeration, entice people to read your manuscript. It should not be redundant with the title and should not contain acronyms or abbreviations. For examples, view our author guidelines: https://journals.plos.org/plosbiology/s/revising-your-manuscript#loc-blurb

Reviewer's comments

Rev. 1: Venkatesan Sundaresan - this reviewer has waived anonymity

The authors have satisfactorily addressed all my comments in the previous review.

Rev. 3:

I thank the authors for their thorough revision of the manuscript that has been much improved. Almost all of my points have been addressed with only a very few remaining: (1) I appreciate the authors' efforts to obtain lines expressing MYB98pro::GFP, MYB98pro::NLS-mRuby2 in myb98 mutant and lines expressing MYB98pro::GFP, MYB98pro::NLS-mRuby2 in Col. wild type, but I do not think the current lack of these lines should prevent publication and cause delays. (2) Figure numbers have been replaced with question marks in this current version of the manuscript. (3) Page 6, line 50: The GO analysis can be improved. For one, it is not clear how the list of genes underlying the GO analysis was obtained. Which samples exactly were compared to get to the "DEG in EC" list, for example? Secondly, the results of the GO analysis are not really discussed. This might be due to the limited amount of meaningful GO terms identified. The author's might want to give this tool a try instead: https://biit.cs.ut.ee/gprofiler/gost.

---

## [Editor Report · Decision Letter 3]

26 Jan 2021

Dear Dr Kurihara,

Thank you for submitting your revised Research Article entitled "Dynamics of the cell fate specifications during female gametophyte development in Arabidopsis" for publication in PLOS Biology.

We are now almost satisfied with the manuscript, but we need you to address a few minor requests before accepting it for production:

- Please indicate in all the corresponding figure legends (including the supplementary ones) where the data can be found.

- In the S1_Data file, the label in Fig. 6D should be 6C. Please correct it.

- Please indicate to what figure corresponds the data shown in the tab labeled as Fig S3D. As far as I can see, there is no section D in Fig. S3.

We expect to receive your revised manuscript within one week.

To submit your revision, please go to https://www.editorialmanager.com/pbiology/ and log in as an Author. Click the link labelled 'Submissions Needing Revision' to find your submission record.

Your revised submission must include the following:

-  a cover letter that should detail your responses to the remaining editorial requests.

Sincerely,

Ines

--

Ines Alvarez-Garcia, PhD,

Senior Editor,

PLOS Biology

---

## [Editor Report · Decision Letter 4]

29 Jan 2021

Dear Dr Kurihara,

On behalf of my colleagues and the Academic Editor, Mark Estelle, I am pleased to say that we can in principle offer to publish your Research Article entitled "Dynamics of the cell fate specifications during female gametophyte development in Arabidopsis" in PLOS Biology, provided you address any remaining formatting and reporting issues. These will be detailed in an email that will follow this letter and that you will usually receive within 2-3 business days, during which time no action is required from you. Please note that we will not be able to formally accept your manuscript and schedule it for publication until you have made the required changes.

PRESS

Thank you again for supporting Open Access publishing. We look forward to publishing your paper in PLOS Biology. 

Sincerely, 

Ines

--

Ines Alvarez-Garcia, PhD 

Senior Editor 

PLOS Biology
